# Soil Enzyme Activity Response under the Amendment of Different Types of Biochar

Piotr Wojewódzki [1] , Joanna Lemanowicz [1] , Bozena Debska [1,* ] and Samir A. Haddad [2] 

[1] Department of Biogeochemistry and Soil Science, Bydgoszcz University of Science and Technology, 6/8 Bernardynska Street, 85-029 Bydgoszcz, Poland; piotr.wojewodzki@pbs.edu.pl (P.W.); jl09@interia.pl (J.L.)

[2] Department of Agricultural Microbiology, Minia University, El-Minia 61517, Egypt; samir.mohamed@mu.edu.eg

[*] Correspondence: bozena.debska@pbs.edu.pl; Tel.: +48-52-374-9502

**Abstract:** Biochar (BC) is a material that finds many applications in agriculture and environmental activities. The aim of the study was to define the influence of biochar produced from various organic materials: mellow compost (MC), stabilized municipal sewage sludge (MSS), pine sawdust (PS), sycamore sawdust (SS) and oak leaves (OL) on soil enzyme activity, as well as its relations with carbon and nitrogen content. After a 60-day incubation of soil and BC, the activity of dehydrogenases (DEH), catalase (CAT), alkaline (AlP) and acid (AcP) phosphatases was investigated. The basic parameters of soil were also determined: TOC, TN, DOM, pH in $H_2O$, available phosphorus (AP). The highest AP content was obtained in the S + MSS, S + OL and S + MC variants. Enzyme activity was highest in soil with MSS BC, regardless of incubation time. After 60 days, the activity of soil enzymes was inhibited. The obtained results indicate that the response of enzymatic activity to biochar depends on the feedstock material and the incubation time. When using BC as an exogenous matter, it is necessary to determine the TOC/TN ratio. For the very wide range of this parameter, supplemental nitrogen fertilization or mixtures of different biochars should be applied.

**Keywords:** biochar; soil; incubation; enzyme activity



## 1. Introduction

Biochar (BC) is a carbon-rich solid product from biomass pyrolysis [1]. It is obtained due to thermochemical conversion of biomass, which includes pyrolysis, torrefaction, gasification and hydrothermal processing [2]. Biochar fabricated on the base of a different kind of biomass increasingly finds more and more application areas. Yang and Ali [3] indicate that biochar is widely used in agricultural and environmental activities. For instance, there are reports about addition of biochar into soils to reduce the leaching of soil nitrogen and phosphorus, soil's heavy-metal pollution, soil's release of an amount of $CO_2$, as well as to improve the photosynthetic physiological characteristics, microbial structure and diversity, crop yield and soil enzyme activities [3]. It was observed that even the addition of small amount of biochar into soil (0.4% and 2.0%) can affect the growth and enzyme activity of soil-resident ligninolytic fungi, promoting their growth and manganese peroxidase activity [4]. The research on 5% addition of biochar from cotton husks, swine manure, eucalyptus sawmill, residue and sugarcane filter cake indicates positive influence of BC obtained in low-temperature pyrolysis (400 °C), on volumetric and available water content of soil [5]. The addition of BC acted as a kind of fertilization, increasing the content of nutrients (N, P, K, S, Ca and Mg) in the soil [6]. An increase in electro conductivity (EC) in soil mixed with BC was also noted [6]. The biochar produced from empty fruit bunch, produced at 400 °C, was reported [7] to possess high cation exchange capacity (CEC) value, which makes it suitable for increasing soil fertility, as well as soil quality. The amendment of BC also positively affects the stability of soil aggregates, especially after a long time (6–12 months) after application [8]. A six-year field experiment [9] confirmed that

application of beech and pine wood chips BC ($9$–$70$ Mg·ha$^{-1}$) enhanced soil aggregation, organic carbon content and pH of the soil. A significant increase in organic carbon storage in the pool of particulate organic matter was also observed. It was also revealed that BC affected the <0.053 mm fraction of soil aggregates.

The issues of soil properties and fertility biological factors should be also taken into account [10]. For example, the enzymatic activity of soil produces a significant impact on nutrient bioavailability to the plant, which in turn affects the level of plant production. Soil enzymes are an effective means of appraising soil quality due to their high sensitivity and the rapid responses elicited to changes in the soil environment [11]. The research on grassland soils [12] revealed links between extracellular enzyme activities and the carbon and nitrogen content in soil. The study indicated a positive relation between β-1,4-glucosidase activity and soil carbon content, as well as L-leucine-amino-peptidase, β-1,4-N-acetyl-glucosaminidase activity and soil nitrogen. Among many factors that influence parameters of the soil environment, soil enzymes and their activity take inherent part in the cycles of soil components, especially degradation and transformation of organic matter and nutrients, as well as mineralization [13].

The influence of BC on enzyme activity is still poorly documented. Earlier studies revealed that biochar changed the biological activity of the soil [14]. According to Bailey et al. [15] and Lehmann et al. [14], the ability of biochar to absorb organic and inorganic particles may be a mechanism for the protection of soil enzymes. Studies by Elzobair et al. [16] revealed that 5 wt% biochar amendment to soil reduces water loss and stabilizes certain enzymes (β-xylosidases), even after denaturing stress. The research of Haddad and Lemanowicz [17] indicated that BC addition to soil resulted in a decrease in enzyme inhibition (arginase and urease) by heavy metals ($Cd^{2+}$, $Pb^{2+}$ and $Ni^{2+}$). However, the results presented by Bailey et al. [15] concerning the relationship between BC and soil enzyme activity did not demonstrate similar trends. Despite the growing interest in BC applications, the processes and mechanisms of BC's long-term impact on the environment have not yet been sufficiently understood. BC properties are different depending on feedstock material [1,4,7]. It may be expected that the effect of BC amendment on soil properties and soil enzyme activity could vary widely. The aim of the study was to define the influence of biochar produced from various organic materials (mellow compost, stabilized municipal sewage sludge, pine sawdust, sycamore sawdust, oak leaves) on soil enzyme activity, as well as its relations with soil carbon and nitrogen content.

## 2. Materials and Methods

### 2.1. Materials and Experimental Setup

The research analyzed the effect of five types of biochar produced from biomass: mellow compost (MC), stabilized municipal sewage sludge (MSS), pine sawdust (PS), sycamore sawdust (SS) and oak leaves (OL). The biochars were divided into two groups, including BC obtained from natural biomass (PS, SS, OL) and waste derived biomass (MSS and MC). However, both groups had waste status produced in sewage treatment plant (MSS), composting plant (MC), sawmill (PS, SS) and green area maintenance (OL).

The biochar was fabricated in the process of low-temperature pyrolysis (400 °C) of air-dried feedstock under atmospheric pressure. Process execution time, after reaching operating temperature, was 60 min. Pyrolysis was carried out in muffle furnace Czylok FCF 22 M (Jastrzębie Zdrój, Poland). The biochar was ground and homogenized in ball mill Retsch PM100 (Haan, Germany). Parameters of milling process: time 4 min, 400 rpm, nine zirconium balls in milling chamber, 30% power of the apparatus. The content of total organic carbon (TOC), total nitrogen (TN), TOC/TN ratio and pH in $H_2O$ in biochars used in the experiment are presented in Table 1.

A small-scale pot experiment was set up to investigate the effect of biochar on soil enzymatic activity. The factor of the experiment was a type of biochar, including control pots with incubated soil without the addition of biochar. One experimental factor on 6 levels was assumed:

- S—soil without biochar;
- S + MC—soil mixed with mellow compost BC;
- S + MSS—soil mixed with sewage sludge BC;
- S + PS—soil mixed with pine sawdust BC;
- S + SS—soil mixed with sycamore sawdust BC;
- S + OL—soil mixed with oak leaves BC.

**Table 1.** Content of total organic carbon (TOC), total nitrogen (TN) and TOC/TN ratio in biochars and soil.

| Sample * | TOC (g kg$^{-1}$) | TN (g kg$^{-1}$) | TOC/TN | pH in H$_2$O |
|---|---|---|---|---|
| MC | 164.8 ± 11.3 | 16.3 ± 3.58 | 10.1 | 9.25 |
| MSS | 371.3 ± 31.3 | 62.9 ± 2.73 | 5.9 | 6.65 |
| PS | 642.2 ± 37.6 | 1.30 ± 0.16 | 494.0 | 6.20 |
| SS | 732.5 ± 42.52 | 2.10 ± 0.28 | 340.7 | 4.00 |
| OL | 594.4 ± 29.4 | 24.0 ± 3.55 | 24.8 | 8.30 |
| S | 11.5 ± 0.93 | 1.42 ± 0.05 | 8.1 | 6.40 |

* S—soil, biochars: MC—mellow compost, MSS—stabilized sewage sludge, PS—pine sawdust, SS—sycamore sawdust, OL—oak leaves.

The evaluation of soil enzyme activity was carried out, taking into account the second factor of the experiment, which was the incubation time.

The incubation pots were filled with mixture of BC and soil in the ratio of dry mass 1:10. Incubation was carried out in the period of 60 days. The studied experiment was set up in triplicate for each incubation period (5, 10, 30, 60 days). The pots were kept in a thermostatic incubator Q-Cell, Poll Lab (Bielsko Biała, Poland). The incubation temperature was 20 ± 2 °C. The soil moisture content was controlled every day. If needed, soil in pots was watered (tap water) to keep the moisture level of 13.5%.

For the incubation experiment, the arable field topsoil material (0–25 cm) was used. According to the WRB classification [18], the sampled soil was classified as Luvisol. It was characterized by the following content of particle-size fractions: sand (2.0–0.05 mm) 50.06%, silt (0.05–0.002 mm) 43.77% and clay (<0.002 mm) 6.18%. According to USDA classification [19], the soil material was sandy loam. The average humidity of the sampled soil was 13.02%, the bulk density was 1.67 g cm$^{-3}$, pH in H$_2$O was 6.40, the content of available phosphorus (AP) was 152 mg kg$^{-1}$. The content of total organic carbon TOC, total nitrogen (TN) and TOC/TN ratio in soil used in the experiment are presented in Table 1.

*2.2. Analysis*

2.2.1. Physico-Chemical Properties

- The contents of total organic carbon (TOC) and total nitrogen (TN) were assayed with the Vario Max CN analyzer provided by company Elementar (Langenselbold, Germany). The content of TOC and Nt was expressed in g kg$^{-1}$ of d.m. of soil.
- The content of dissolved organic carbon (DOC) and dissolved nitrogen (DTN), in samples after incubation, were assayed in the solutions from the extraction of soil samples of 0.004 mol dm$^{-3}$ CaCl$_2$, at the ratio of soil sample: extractant of 1:50. Extraction took 1 h, and then, the solution was centrifuged. The content of DOC and DTN was assayed with Multi N/C 3100 Analityk Jena (Jena, Germany) analyzer and expressed in mg kg$^{-1}$ d.m. of the soil sample, as well as the percentage share in the pool: TOC and TN, respectively.
- pH in H$_2$O was determined with a potentiometric method according to the standard PN-ISO 10390 [20].
- The content of available forms of phosphorus (AP) in the samples after incubation was analyzed according to the standard PN-R-04023 [21].
- Soil particle-size distribution was determined with Malvern Instruments Mastersizer 2000 analyzer (Malvern, UK), equipped with dispersing device Hydro 2000MU.

- Soil dry mass and humidity were tested with weigh-dryer MAC-50 NH Radwag (Radom, Poland).

The soil samples prior analyses of TOC, TN, DOC, DTN, pH and AP were air dried and powdered.

### 2.2.2. Enzyme Analysis

Enzyme activity analyses were performed in fresh soils, sieved on 2 mm screens, that had been stored at 4 °C. The activity of soil enzymes was tested at the beginning of the experiment and then after 5, 10, 30 and 60 days.

- The activity of dehydrogenases (DEH) was assayed with consideration of Thalmann protocol [22], with sample incubation with 2,3,5-triphenyltetrazolium chloride and measurement of the absorbance of triphenylformazane (TPF) at 546 nm. The results were presented in mg TPF kg$^{-1}$ 24 h$^{-1}$.
- The activity of catalase (CAT) was determined with the Johnson and Temple method [23], with 0.3% hydrogen peroxide solution as a substrate. Residual $H_2O_2$ was determined by titration with 0.02 M $KMnO_4$ under acidic conditions.
- The activity of soil alkaline phosphatase (AlP) and acid phosphatase (AcP) was measured on the ground of detection of p-nitrophenol (pNP) released after incubation (37 °C, 1 h) at pH ~ 6.5 for AcP and pH ~ 11.0 for AlP [24].

According to obtained results of each enzyme activity, the geometric mean of enzyme activities (G*Mea*) [25] was also calculated:

$$\mathrm{G}Mea = \sqrt[4]{(\mathrm{CAT \cdot DEH \cdot AlP \cdot AcP})} \tag{1}$$

where DEH, CAT, AlP, AcP are dehydrogenases, catalase, alkaline phosphatase, acid phosphatase, respectively.

The resistance index (*RS*) was determined on the basis of enzyme activity, according to Orwin and Wardel's [26] formula:

$$RS = 1 - \left[ \frac{2|\mathrm{D0}|}{\mathrm{C0} + |\mathrm{D0}|} \right] \tag{2}$$

where D0 = C0 − P0, C0 enzyme activity in reference soil, P0—enzyme activity in soil after BC addition. The value of *RS* ranges from −1 to +1.

### 2.2.3. Statistical Analysis

The experimental results were statistically analyzed using the Statistica software (StatSoft, Kraków, Poland). Data concerning enzyme activity (DEH, CAT, AlP, AcP) were analyzed using the two-way ANOVA, where the first factor was the type of BC and the second factor was the incubation time. The other parameters were analyzed in one-way ANOVA. Using the analysis of variance, the strength of the $\eta^2$ effect was determined, which indicates the percentage share of qualitative variables in the development of enzyme activity in the soil.

Statistical data analysis also concerned the principal component analysis (PCA method) and the cluster analysis (CA method). The PCA and CA analyses are statistical multivariate methods commonly applied in environmental research, allowing the determination of groups with similar characteristics. These methods are not equivalent, only complementary. Principal component analysis allows for reducing the number of variables describing a given object, as well as indicating the influence of primary variables on principal components and the mutual correlations of primary variables. In the presented research, there are 11 variables (pH in $H_2O$, TOC, TN, TOC/TN, DOC, DTN, AP, DEH, CAT, AlP, AcP) describing each of the tested samples after 60 days of incubation. The cluster analysis allows for the separation of groups of objects based on the differentiation of variables. Agglomeration of properties was assessed by Ward's cluster analysis with Euclidean dis-

tance [27] concerning the following parameters: pH in $H_2O$, TN, DOC, DTN, AP, DEH, CAT, AlP, AcP. Because the obtained results on chemical and enzymatic properties of the soil were expressed in different units, the principal components were calculated using the correlation matrix.

All analyses were performed in triplicate.

## 3. Results and Discussion

The biochar used in the experiment differed in the content of TOC and TN (Table 1). The highest TOC content was found in biochar obtained from natural biomass: sycamore sawdust (732.5 g kg$^{-1}$), pine sawdust (642.2 g kg$^{-1}$) and oak leaves (594.4 g kg$^{-1}$). The second group of biochars were biochars obtained from compost and sewage sludge, characterized by lower carbon content (164.8 and 371.3 g kg$^{-1}$, respectively) and higher nitrogen content than biochars obtained from sawdust and leaves. The lower nitrogen content in wood than in leaves resulted in its minor contribution to sawdust biochar (SS, PS). The lowest value of the TOC/TN ratio—5.9 was characteristic for biochar obtained from MSS, and the highest value was calculated for PS biochar (494.0). The feedstock of organic material for biochar fabrication determined its pH values, ranging from 4.00 (SS BC) to 9.25 (MC BC).

The content of TOC and TN in the BC determined the content of these elements and the value of the TOC/TN ratio in the samples subjected to incubation (Table 2). The data presented in Table 2 indicate that the lowest TOC/TN value was characteristic for the sample obtained after mixing soil with MSS BC. The values of the discussed ratio obtained for the S + MC variant was the closest to the TOC/TN value of soil without additives. TOC/TN values are one of the parameters determining the intensity of organic matter decomposition. The highest TOC/TN values were noted for S + SS and S + PS variants.

**Table 2.** Content of total organic carbon (TOC), total nitrogen (TN) and TOC/TN ratio in soil mixed with biochar before (0) and after 60 days of incubation (1).

| Variant * | $TOC_0$ (g kg$^{-1}$) | $TOC_1$ (g kg$^{-1}$) | $TN_0$ (g kg$^{-1}$) | $TN_1$ (g kg$^{-1}$) | $TOC_0/TN_0$ | $TOC_1/TN_1$ |
|---|---|---|---|---|---|---|
| S | 11.50 | 11.48 [f] ** ± 0.89 | 1.42 | 1.40 [d] ± 0.04 | 8.10 | 8.20 |
| S + MC | 26.83 | 24.27 [e] ± 0.93 | 2.91 | 2.52 [c] ± 0.30 | 9.22 | 9.63 |
| S + MSS | 47.48 | 45.08 [d] ± 2.00 | 7.57 | 7.07 [a] ± 0.77 | 6.27 | 6.38 |
| S + PS | 74.59 | 74.10 [b] ± 4.44 | 1.41 | 1.12 [e] ± 0.19 | 52.90 | 66.16 |
| S + SS | 83.60 | 83.50 [a] ± 2.87 | 1.49 | 1.23 [de] ± 0.16 | 56.11 | 67.89 |
| S + OL | 69.79 | 66.94 [c] ± 3.26 | 3.68 | 3.52 [b] ± 0.05 | 18.96 | 19.02 |

* S—soil without biochar, S + MC—soil with mellow compost, S + MSS—soil with stabilized municipal sewage sludge, S + PS—soil with pine sawdust, S + SS—soil with sycamore sawdust, S + OL—soil with oak leaves. ** Values followed by the same letters are not significantly different at 5%.

In the incubation pot experiment, a significant decrease in TOC content was noted in variants S + MC, S + MSS and S + OL (Figure 1A). Those biochars were materials with a smaller range of TOC/TN ratio compared to PS and SS biochar. The loss of TOC in the samples ranged from 4.08% (variant S + OL) to 9.54% (variant S + MC) (Figure 1B). The nitrogen content in the samples after incubation was lower in comparison to initial state by 0.161 g kg$^{-1}$ (S + OL) to 0.499 g kg$^{-1}$ (S + MSS), which was 4.37% and 13.27%, respectively. The greatest loss of nitrogen in relation to its initial content was noted for variants containing PS biochar (20.47%) and SS biochar (17.59%). In absolute values, it amounted to 0.288 g kg$^{-1}$ and 0.263 g kg$^{-1}$, respectively.

For the variants S + PS and S + SS, a wider range of the TOC/TN ratio was noted. It is a consequence of greater nitrogen losses in comparison to carbon content decrease. The above dependencies may indicate differences in the structure of the BC used. The PS and SS

biochars are most likely characterized by a more condensed structure of aromatic rings and a low proportion of nitrogen-containing aliphatic structures. According to Copper et al. [9], the structure of biochar is determined primarily by the chemical composition of the material subjected to pyrolysis. It also depends on the process temperature.

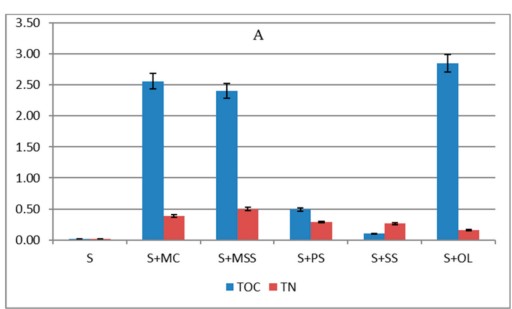 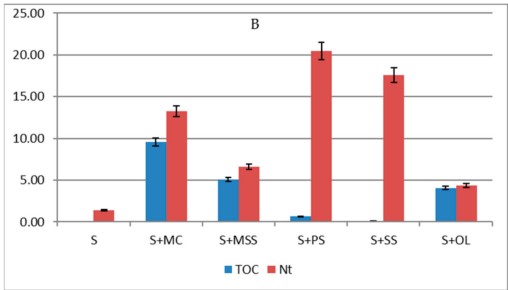

**Figure 1.** Changes in TOC and TN content between the initial and final incubation stages expressed in g $kg^{-1}$ (**A**) and % (**B**).

The most mobile and rapidly decomposing fraction of organic matter (OM) is the so-called dissolved organic matter (DOM), the content of which is determined based on the carbon and nitrogen concentration in water extracts—dissolved organic carbon (DOC) and dissolved nitrogen (DTN). The DOC in arable soils generally accounts for less than 1% of the TOC. Despite such a small share, the DOM has a major role in the carbon and nitrogen biogeochemical cycle and can be a source of nutrients for microorganisms [28,29]. Generally, it is assumed that DOC content alteration can be an important indicator of changes that occur in soils, especially due to anthropogenic factors [28], which also include fertilization with exogenous organic matter, including biochar. The soil sample containing MSS biochar was characterized by the highest content of DOC and DTN (Table 3). In each case, it was observed that the addition of BC to soil increased the DOC content. The increase in DTN content was found only after the introduction of MC and MSS biochars to the soil. Despite the increase in the content of DOC in the soil after the addition of biochar, it did not increase the share of this fraction of organic matter. The contribution of DOC in the S + MC variant was lower by 0.86 percentage points than its contribution in soil (1.91%). Additionally, in comparison to soil (S), the participation of DOC in variant S + MSS was lower by 0.56%; in variants with SS, PS, OL biochars, it was lower, in the range of 1.59 to 1.39 percentage points. In comparison to soil without biochar amendment (DTN = 3.25%), the decrease in DTN in analyzed variants with biochar ranged from 2.11 (S + OL) to 0.9 percentage points (variant S + MC). The one exception was S + MSS variant, where the DNT increased by 1.71 percentage points.

The results of the pH in $H_2O$ measurement after 60 days of incubation ranged from a value of 5.75 to 7.04 and depended on the type of added biochar (Table 3). S + PS, S + SS samples were acidic, (S) and S + MSS slightly acidic, S + MC acidic and S + OL basic.

The AP content ranged from 102 to 148 mg $kg^{-1}$, which, according to the standard PN-R-04023 [21], classifies it as soils with a very high content of this macronutrient (class I) (Table 3). Due to the environmental aspect, the optimal concentration of phosphorus in the soil should be in the class of medium abundance (45–66 mg P $kg^{-1}$). The applied experimental factor (BC) significantly determined the AP content in soil.

There was no significant difference in AP concentration between S + MSS, S + OL and S + MC, where its content was the highest. The BC into soil application resulted in AP content reduction in soil samples compared to the content before the experiment. According to Glaser and Lehr [30], the addition of biochar increases the AP content in agricultural soil, regardless of the feedstock material used for the production of biochar. However, the increase in AP content was dependent on the soil pH. Generally, it should be considered that the soil before the experiment was characterized by a very high content of phosphorus—152.0 mg $kg^{-1}$.

**Table 3.** Values of pH in $H_2O$, content of available phosphorus (AP), content and share of dissolved organic carbon (DOC) and nitrogen DTN (after 60-day incubation).

| Variant * | pH in $H_2O$ | AP (mg kg$^{-1}$) | DOC (mg kg$^{-1}$) | DOC (%) | DTN (mg kg$^{-1}$) | DTN (%) |
|---|---|---|---|---|---|---|
| S | 6.49 | 130 [b] ** $\pm$ 0.7 | 219.5 [f] $\pm$ 9.5 | 1.91 [a] $\pm$ 0.12 | 46.0 [c] $\pm$ 4.09 | 3.25 [b] $\pm$ 0.36 |
| S + MC | 7.04 | 145 [a] $\pm$ 1.8 | 255.5 [e] $\pm$ 6.1 | 1.05 [c] $\pm$ 0.02 | 59.3 [b] $\pm$ 2.38 | 2.35 [c] $\pm$ 0.32 |
| S + MSS | 6.24 | 148 [a] $\pm$ 1.1 | 606.8 [a] $\pm$ 13.2 | 1.35 [b] $\pm$ 0.04 | 350.8 [a] $\pm$ 10.13 | 4.96 [a] $\pm$ 0.63 |
| S + PS | 5.75 | 102 [d] $\pm$ 4.2 | 301.0 [c] $\pm$ 8.7 | 0.41 [e] $\pm$ 0.03 | 16.4 [e] $\pm$ 0.53 | 1.46 [e] $\pm$ 0.19 |
| S + SS | 5.84 | 121 [c] $\pm$ 3.8 | 274.8 [d] $\pm$ 7.5 | 0.33 [f] $\pm$ 0.02 | 24.5 [d] $\pm$ 3.06 | 1.99 [d] $\pm$ 0.15 |
| S + OL | 7.44 | 146 [a] $\pm$ 4.0 | 349.5 [b] $\pm$ 18.7 | 0.52 [d] $\pm$ 0.04 | 39.9 [c] $\pm$ 2.34 | 1.14 [e] $\pm$ 0.07 |

* S—soil without biochar, S + MC—soil with mellow compost, S + MSS—soil with stabilized municipal sewage sludge, S + PS—soil with pine sawdust, S + SS—soil with sycamore sawdust, S + OL—soil with oak leaves. ** Values followed by the same letters are not significantly different at 5%.

The DEH activity dependence on the type of BC and the incubation time are presented in Table 4. DEH activity was the highest after 30 days of incubation in soil samples with MSS (1.529 mg TPF kg$^{-1}$ 24 h$^{-1}$), PS (1.231 mg TPF kg$^{-1}$ 24 h$^{-1}$) and SS (0.651 mg TPF kg$^{-1}$ 24 h$^{-1}$). The addition of biochar to the soil resulted in a decrease in dehydrogenase activity compared to the control, excluding the S + MSS variant. The lowest DEH activity was observed after 60 days of incubation (except for the soil with OL biochar addition). BC provides the soil with reducing conditions that enhance electron reduction and accelerate the activity of dehydrogenase [31]. Dehydrogenases [EC 1.1.1] are the main representatives of the class of oxidoreductase enzymes. Their activity can be considered a good indicator of oxidative metabolism in soil and thus microbial activity. The role of dehydrogenases is related to many biochemical processes in soil, including the emission of greenhouse gases $CO_2$ and $N_2O$.

**Table 4.** The activity of dehydrogenase DEH (mg TPF kg$^{-1}$ 24 h$^{-1}$).

| Variant * | Time of Incubation | | | |
|---|---|---|---|---|
| | 5 Days | 10 Days | 30 Days | 60 Days |
| S | 1.09 [bC] ** $\pm$ 0.01 | 1.21 [bA] $\pm$ 0.010 | 1.16 [bB] $\pm$ 0.038 | 0.93 [bD] $\pm$ 0.017 |
| S + MC | 0.87 [dAB] $\pm$ 0.0 | 0.90 [cA] $\pm$ 0.007 | 0.82 [cB] $\pm$ 0.005 | 0.75 [cC] $\pm$ 0.018 |
| S + MSS | 1.38 [aB] $\pm$ 0.028 | 1.44 [aB] $\pm$ 0.023 | 1.53 [aA] $\pm$ 0.026 | 1.30 [aC] $\pm$ 0.010 |
| S + PS | 0.95 [cC] $\pm$ 0.011 | 1.0 [cB] $\pm$ 0.054 | 1.23 [bA] $\pm$ 0.069 | 0.79 [cD] $\pm$ 0.029 |
| S + SS | 0.64 [eAB] $\pm$ 0.02 | 0.59 [dB] $\pm$ 0.008 | 0.65 [dA] $\pm$ 0.029 | 0.55 [dB] $\pm$ 0.019 |
| S + OL | 0.46 [fA] $\pm$ 0.035 | 0.40 [eAB] $\pm$ 0.01 | 0.29 [eB] $\pm$ 0.013 | 0.44 [eA] $\pm$ 0.023 |

$\eta^2$ for variant of biochar 91.93%, $\eta^2$ for time of incubation 3.14%, $\eta^2$ for interaction 4.36%, $\eta^2$ for error 0.534%

* S—soil without biochar, S + MC—soil with mellow compost, S + MSS—soil with stabilized municipal sewage sludge, S + PS—soil with pine sawdust, S + SS—soil with sycamore sawdust, S + OL—soil with oak leaves; ** Different small letters indicate significant differences among variants. Different capital letters indicate significant differences among incubation periods.

CAT activity was highest in soil with the addition of biochar from sewage sludge (S + MSS), regardless of the incubation time (Table 5). Catalase is a cellular antioxidant enzyme that protects against oxidative stress. It catalyzes the decomposition of hydrogen peroxide into water and oxygen. Catalase activity is used together with dehydrogenase activity to obtain information about microbial activity in soil [32]. CAT activities were greatest after 30 days of incubation in the following variants: S + MSS, S + PS, S + SS and S + OL. For the control soil (S) and in variant S + MC, its activity was highest after 10 days of incubation.

The dynamics of AlP and AcP activity changes caused by BCs in the experimental period are presented in Tables 6 and 7. Acid (EC 3.1.3.2) and alkaline phosphatase (EC 3.1.3.1) are the most frequently studied soil enzymes because they react rapidly to environmental stress caused by anthropogenic and natural factors, especially to changes in soil pH. The

acidic pH, in the range of 4–6, is optimal for acid phosphatase, and alkalic (pH 8–10) for alkaline phosphatase. The highest activity of AlP and AcP was found in the variant S + MSS. It was higher than the activity of both phosphatases in soil (S) by 200% and 3%, respectively.

**Table 5.** The activity of catalase CAT (mg $H_2O_2$ kg$^{-1}$ h$^{-1}$).

| Variant * | Time of Incubation | | | |
|---|---|---|---|---|
| | 5 Days | 10 Days | 30 Days | 60 Days |
| S | 0.92 [bB] ** ± 0.013 | 1.08 [bA] ± 0.041 | 0.94 [bB] ± 0.013 | 0.82 [bC] ± 0.014 |
| S + MC | 0.63 [dC] ± 0.015 | 0.81 [cA] ± 0.006 | 0.74 [dB] ± 0.021 | 0.62 [dC] ± 0.010 |
| S + MSS | 1.28 [aC] ± 0.024 | 1.35 [aB] ± 0.025 | 1.45 [aA] ± 0.027 | 1.01 [aD] ± 0.072 |
| S + PS | 0.75 [cC] ± 0.027 | 0.85 [cB] ± 0.023 | 0.92 [bA] ± 0.010 | 0.72 [cC] ± 0.009 |
| S + SS | 0.54 [eD] ± 0.029 | 0.71 [dB] ± 0.012 | 0.82 [cA] ± 0.013 | 0.65 [dC] ± 0.025 |
| S + OL | 0.43 [fC] ± 0.024 | 0.55 [eB] ± 0.027 | 0.64 [cA] ± 0.023 | 0.54 [eB] ± 0.020 |

$\eta^2$ for variant of biochar 83.33%, $\eta^2$ for time of incubation 10.21%, $\eta^2$ for interaction 5.48%, $\eta^2$ for error 0.976%

\* S—soil without biochar, S + MC—soil with mellow compost, S + MSS—soil with stabilized municipal sewage sludge, S + PS—soil with pine sawdust, S + SS—soil with sycamore sawdust, S + OL—soil with oak leaves; ** Different small letters indicate significant differences among variants. Different capital letters indicate significant differences among incubation periods.

**Table 6.** The activity of alkaline phosphatase AlP (mMpNP kg$^{-1}$ h$^{-1}$).

| Variant * | Time of Incubation | | | |
|---|---|---|---|---|
| | 5 Days | 10 Days | 30 Days | 60 Days |
| S | 0.90 [bcA] ** ± 0.02 | 0.95 [bcA] ± 0.069 | 0.73 [cB] ± 0.029 | 0.47 [cC] ± 0.024 |
| S + MC | 0.87 [cB] ± 0.024 | 1.02 [bA] ± 0.044 | 0.90 [bB] ± 0.074 | 0.61 [bC] ± 0.020 |
| S + MSS | 1.66 [aB] ± 0.089 | 1.33 [aC] ± 0.031 | 1.82 [aA] ± 0.056 | 1.31 [aC] ± 0.017 |
| S + PS | 0.94 [bA] ± 0.29 | 0.90 [cAB] ± 0.009 | 0.84 [bB] ± 0.015 | 0.40 [cC] ± 0.010 |
| S + SS | 0.69 [eA] ± 0.03 | 0.61 [dB] ± 0.025 | 0.65 [cAB] ± 0.06 | 0.30 [dC] ± 0.012 |
| S + OL | 0.74 [dC] ± 0.05 | 0.82 [cB] ± 0.020 | 0.89 [bA] ± 0.023 | 0.55 [bcD] ± 0.010 |

$\eta^2$ for variant of biochar 72.33%, $\eta^2$ for time of incubation 17.05% $\eta^2$ for interaction 6.89%, $\eta^2$ for error 3.733%

\* S—soil without biochar, S + MC—soil with mellow compost, S + MSS—soil with stabilized municipal sewage sludge, S + PS—soil with pine sawdust, S + SS—soil with sycamore sawdust, S + OL—soil with oak leaves; ** Different small letters indicate significant differences among variants. Different capital letters indicate significant differences among incubation periods.

The addition of the remaining biochars to the soil reduced the AcP activity, whereas the activity of AlP in variants S + PS and S + SS was lower compared to S. The AlP activity increased in variants S + MC and S + OL. The incubation time significantly influenced the dynamics of the activity of both phosphatases, which varied depending on the feedstock biomass used for BC fabrication. According to Elzobair et al. [16] and Foster et al. [33], the surface of biochar has a high absorption potential, including enzymes, thus changing their activity by changing the conformation in the active site of the enzyme or rotating the active site toward the surface of the biochar. Therefore, the important factors determining the enzymatic activity of soil after adding biochar from various materials could be the BC surface area and pore size. Those parameters depend on feedstock biomass used for BC production. If the enzyme activity is reduced, the circulation of carbon and nutrients in the soil may be slowed down.

The analysis of the $\eta^2$ coefficients indicated that the type of biochar explains the variability of DEH activity by 92%, of CAT by 83.5%, of AlP by 72.33% and of AcP by 91% (Tables 4–7). However, the incubation time only influenced the variability of DEH by 4.35%, CAT by 10.21%, AlP by 17.05% and AcP by 4.9%.

Soil processes are determined by their physical, chemical and biochemical properties. Changes in soil properties can be presented, inter alia, by using index of soil quality. One of them is the geometric mean G*Mea* [25]. On average, for the variant S + MSS, a 38% G*Mea* increase was noted in comparison to the control (S) (Figure 2). For other variants, a decrease

in GMea was observed: by 24% (S + MC), 11% (S + PS), 41% (S + SS) and 49% (S + OL). The lowest GMea value was specified after 60 days of incubation (mean GMea = 0.849). The studies of Paz-Ferreiro et al. [34] revealed lower variability of GMea values in comparison to the variability of the activity of individual enzymes (phosphomonoesterase, β-glucosidase, arylsulphatase, urease, invertase and β-glucosaminidase). This suggests that GMea is a more appropriate indicator for assessing soil quality than a single soil enzyme activity. According to Jing et al. [35], the GMea index is an appropriate tool to estimate the influence of biochar on the activity of soil enzymes. The authors observed that the added rice straw biochar increased the GMea value. They also indicated that it is beneficial for improving the mineralization of organic matter and soil fertility.

**Table 7.** The activity of acid phosphatase AcP (mMpNP kg$^{-1}$ h$^{-1}$).

| Variant * | Time of Incubation | | | |
|---|---|---|---|---|
| | 5 Days | 10 Days | 30 Days | 60 Days |
| S | 2.86 bC **± 0.05 | 3.09 aA ± 0.099 | 3.01 bB ± 0.103 | 2.53 aD ± 0.050 |
| S + MC | 1.83 dA ± 0.06 | 1.66 cB ± 0.148 | 1.40 dC ± 0.083 | 1.42 dC ± 0.026 |
| S + MSS | 3.06 aB ± 0.03 | 3.17 aA ± 0.050 | 3.21 aA ± 0.034 | 2.41 bC ± 0.032 |
| S + PS | 2.41 cB ± 0.03 | 2.35 bB ± 0.061 | 2.58 cA ± 0.160 | 1.81 cC ± 0.012 |
| S + SS | 1.44 eA ± 0.04 | 1.18 dC ± 0.036 | 1.29 eB ± 0.049 | 1.14 eC ± 0.025 |
| S + OL | 1.10 fB ± 0.05 | 1.11 dB ± 0.037 | 0.74 fC ± 0.030 | 1.21 fA ± 0.016 |

$\eta^2$ for variant of biochar 91.03%, $\eta^2$ for time of incubation 3.417% $\eta^2$ for interaction 4.89%, $\eta^2$ for error 0.665%

* S—soil without biochar, S + MC—soil with mellow compost, S + MSS—soil with stabilized municipal sewage sludge, S + PS—soil with pine sawdust, S + SS—soil with sycamore sawdust, S + OL—soil with oak leaves; ** Different small letters indicate significant differences among variants. Different capital letters indicate significant differences among incubation periods.

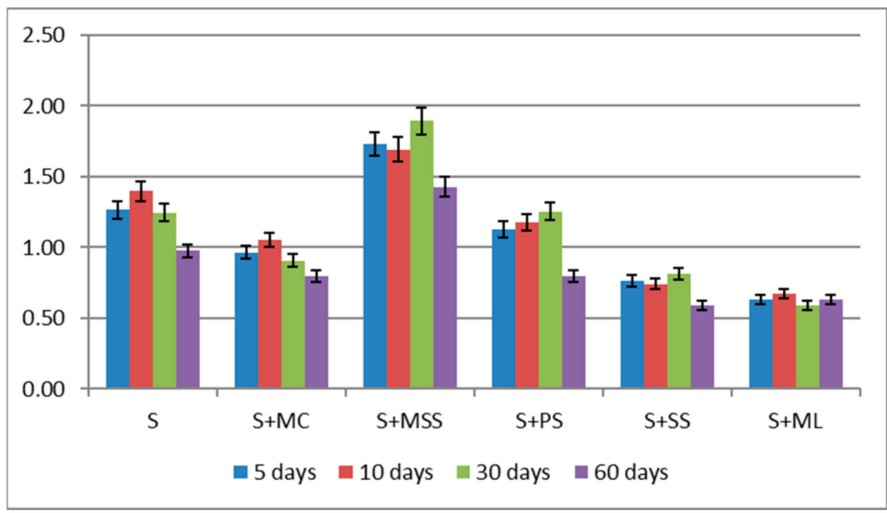

**Figure 2.** Soil quality indicator—the geometric mean GMea.

The *RS* values presented in Figure 3 indicate that the analyzed enzymes varied in their sensitivity to the addition of biochars at different incubation periods. The highest mean value of *RS* was recorded for AlP (0.558), and the lowest for AlP (0.491). Regardless of the type of biochar used, DEH resistance was the highest after 5 days of incubation (average *RS* = 0.538), CAT after 30 days (average *RS* = 0.639), AlP after 10 days (average *RS* = 0.683) and AcP after 5 days (mean *RS* = 0.527). The highest *RS* value was noted in variant S + PS: DEH (average *RS* = 0.789), CAT (average *RS* = 0.684), AlP (average *RS* = 0.746). The highest AcP resistance (average *RS* = 0.900) was observed in variant S + SM. Higher values of the *RS* index indicate that biochar had minor effect (maximum resistance) on soil enzyme activity [26].

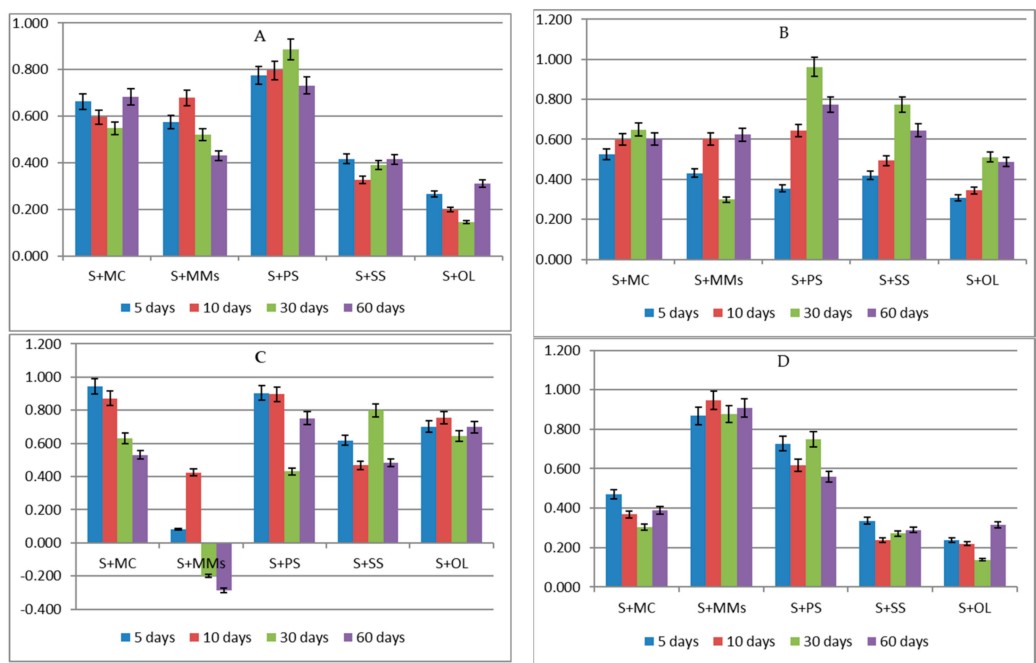

**Figure 3.** Soil resistance indices (*RS*) of soil enzymes: DEH (**A**); CAT (**B**); AlP (**C**); AcP (**D**).

The PCA principal components technique was used to explain the differentiation of the soil in terms of the enzymes tested (DEH, CAT, AlP and AcP), pH in $H_2O$ and the content of AP, TOC, DOC, TN, DTN, TOC/TN after 60-day incubation based on the two main components, PC1 and PC2. The first two principal components describe 82.23% of the total variance of the original data set (Figure 4A). The PCA analysis indicated that the first component (PC1) generated 58.49% of the total variance. PC1 was negatively related to the activity of DEH ($-0.819$), CAT ($-0.762$), AlP ($-0.964$) and AcP ($-0.638$), as well as the content of TN ($-0.904$), DOC ($-0.791$) and DTN ($-0.930$). The second component (PC2) distinguished 23.74% of the total variance and was significantly positively related to the AP content (0.726) and pH in $H_2O$ (0.871). The studied enzymes were grouped on the PC1 side, which, therefore, can be generally equated with the biochars' effect on the soil.

PCA analysis made it possible to verify the importance of mutual correlations among the studied parameters. The TOC content was significantly negatively correlated with AcP activity ($r = -0.608$); TN content was positively correlated with AlP ($r = 0.952$), CAT ($r = 0.580$), DEH ($r = 0.612$) and AP content ($r = 0.686$). A study by Li et al. [36] also presented a positive relationship between the content of nitrogen and the activity of AlP. Guan et al. [37] suggest that nitrogen may stimulate the number of microorganisms, increasing the demand for phosphorus, and this leads to an increase in phosphatase activity. The study [37] also did not indicate positive correlation between TOC and soil enzymes, which was explained by the addition of organic substances of anthropogenic origin into soil. These substances do not act as substrates for the tested enzymes because they do not occur in the natural organic matter of the soil. According to Feng et al. [38], higher enzyme activity could accelerate the decomposition rate of soil organic matter (SOM), leading to the depletion of soil organic carbon (SOC). These authors believe that when the SOC content is low, the activity of the enzymes may be inhibited due to a lack of energy and substrates. This suggests that the enzymatic activity is not a perfect reflection of the SOC content. According to Bielińska et al. [39], the lack of relationship between the content of organic carbon and the enzymatic activity of soils may be related to the low contribution of humic substances in the total content of soil organic matter, which in consequence limits the availability of easily digestible carbon, which determines the growth of soil enzyme-producing bacteria. Enzymes can be bound in humic complexes, which can protect enzyme proteins, but the substrates of high molecular weight may also be a reason

for enzyme deactivation [40]. The results of presented research revealed a relation between the content of DOC and the activity of DEH. A positive correlation (Figure 4A) between the content of DOC and the activity of DEH (*r* = 632), CAT (*r* = 0.651) and AlP (*r* = 834), was noted. Similar results were obtained by Karimi et al. [41]. These authors concluded that the influence of biochar on the activity of catalase and soil dehydrogenase may be related to the increased content of DOC in the soil after biochar application. Presented results (Table 3) indicate that BC application results in DOC content increase in comparison to control soil (variant S). DOC increase ranged from 16.4% (S + MC) to 176.5% (S + MSS). According to Haney et al. [42] and Wang et al. [43], the changes in enzyme activity and DOM content can be used as important indicators of soil quality. Wang et al. [43] also concluded that the relationship between the activity of soil enzymes and the content of labile fractions of organic carbon offers the possibility of a potential assessment of changes in the biochemical cycle of elements in relation to changes in soil use. The content of DTN (similarly to TN) was also positively correlated with DEH (*r* = 0.824), CAT (*r* = 0.813) and AlP (*r* = 0.938). The significant positive relationship was demonstrated between the nitrogen content in soluble organic matter and the content and contribution of total nitrogen (Figure 4A).

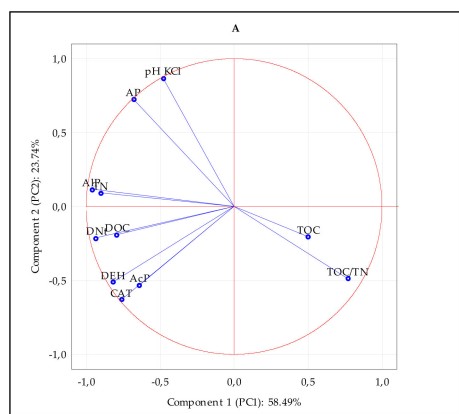 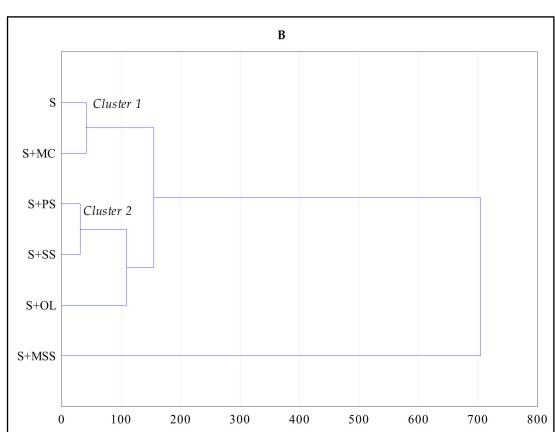

**Figure 4.** Configuration of variables in the system of the first two axes, PC1 and PC2, of principal components (**A**); cluster analysis (**B**).

The sensitivity of enzymes as indicators of changes that occurred in soil under the influence of natural and anthropogenic factors is higher in the case of oxidoreductases than in the case of hydrolases [44]. Principal component analysis indicates that AP content rose with increasing activity of alkaline phosphatase (*r* = 0.755); however, AP and AcP were not correlated. Phosphatases are enzymes whose activity measurement is used to assess the potential rate of phosphorus mineralization in soil [45]. A positive relationship was found between pH in $H_2O$ and AP content (*r* = 0.954) and AlP activity (*r* = 0.599), which corresponds to the finding that phosphomonoesterases are enzymes that are very sensitive to changes in soil pH [46].

The dendrogram of cluster analysis (Figure 4B) illustrates the similarities among different variants of soil mixed with biochar after 60-day incubation. The smaller the Euclidean distance, the more similar the objects. Two main clusters are distinguished in the dendrogram. The first cluster includes the S and S + MC variants with the lowest TOC and DOC content. The second cluster refers to variants S + PS, S + SS and S + OL, where the closest similarity occurs between the variants with sawdust biochar (SS, PS). The variant S + MSS was characterized by the greatest Euclidean distance in relation to the first and second clusters. This variant was identified by the highest content of TN, DOC, DTN, AP, as well as activity of DEH, CAT, AlP and AcP. The study of Hossain et al. [47] on wastewater sludge biochar indicates that this type of BC presents considerable differences in relation to biochars obtained from other biomass of natural origin [1,4,48], which could explain the separate aggregation of S + MSS variant.

## 4. Conclusions

After 6 months of incubation, a slight decrease in carbon content (on average by $2.6\,\mathrm{g\,kg^{-1}}$) was recorded for the soil incubated with biochars obtained from compost, sewage sludge and oak leaves. The sawdust biochar (SS, PS) characterized with wide TOC/TN ratio did not significantly change the carbon content in soil after the incubation. Therefore, it is possible to recommend the use of biochar as a source of exogenous organic matter favoring carbon sequestration.

The obtained results indicate that the response of enzymatic activity to biochar depends on the biochar feedstock and the incubation time. Due to the wide differentiation of soil enzyme activity, depending on the biomass feedstock for BC fabrication and the incubation time, its usefulness for environmental purposes may be different. The enzymatic activity, which is an indicator of soil fertility, increased after MSS biochar addition into the soil. However, the low value of the TOC/TN ratio observed after adding this biochar may increase the intensity of organic matter decomposition. When using biochar as an exogenous matter, it is necessary to determine the TOC/TN ratio. For the very wide range of this parameter, supplemental nitrogen fertilization or mixtures of biochars (for example, sewage sludge BC and wood sawdust BC) should be applied.

**Author Contributions:** Conceptualization, P.W., J.L. and B.D.; methodology, P.W., J.L. and B.D.; investigation, P.W., J.L. and B.D.; data curation—compilation and analysis of results, J.L., B.D. and P.W.; writing—original draft preparation, J.L., P.W. and B.D.; writing—review and editing, B.D., J.L., P.W. and S.A.H. All authors have read and agreed to the published version of the manuscript.

**Funding:** Bydgoszcz University of Science and Technology under Grant BN 31/2019 and 38/2019.

**Institutional Review Board Statement:** Not applicable.

**Informed Consent Statement:** Not applicable.

**Data Availability Statement:** Not applicable.

**Conflicts of Interest:** The authors declare no conflict of interest.

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
