# Peer review of "Soil Enzyme Activity Response under the Amendment of Different Types of Biochar"

_agronomy, doi:10.3390/agronomy12030569_

Round 1
Reviewer 1 Report
Scientific research into the properties of biochar and its transformation in soil is up-to-date and needed. The application of biochar to soil may be one of the most important factors increasing the level of carbon sequestration by soil. The work has good intentions and should be published with minor explanation.
1. Total organic carbon (TOC) content. On what basis was this form of coal considered? Was the total carbon content (TC) determined on the analyzer? Could the inorganic form of carbon (IC) occur in biocarbon, e.g. in the form of CaCO3?
2. What mass of soil was incubated? How many repetitions have the incubation experiment been set up? Could sampling for enzyme tests on days 5, 10 and 30 of incubation have an impact on the course of the experiment?
3. How many replications were laboratory analyzes performed?
Author Response
Dear Reviewer
The paper's authors wish to thank for all the precious comments and guidelines; we have tried to consider all of them.
I hope that the corrections introduced will be satisfactory.
With best regards,
Bozena Debska
Answers:
- Total organic carbon (TOC) content. On what basis was this form of coal considered? Was the total carbon content (TC) determined on the analyzer? Could the inorganic form of carbon (IC) occur in biocarbon, e.g. in the form of CaCO3?
The content of TOC is a result of total carbon TC and inorganic carbon IC determination. Because the content of IC was 0 the value of TC was equal to TOC.
- What mass of soil was incubated? How many repetitions have the incubation experiment been set up? Could sampling for enzyme tests on days 5, 10 and 30 of incubation have an impact on the course of the experiment?
There was set up small scale incubation experiment. Each incubation pot contained mixed BC and soil in the ratio 1:10 (calculated for dry mass). There was used 5 g of BC and 45 g d.m. of soil for each incubation pot. The studied experiment was set up in triplicate for each incubation period. Altogether there was 72 incubation pots – 18 for each periods.
Information on experiment repetition – line 109.
- How many replications were laboratory analyzes performed?
Laboratory analysis were performed in triplicate (information added in point 2.2 of manuscript) – line 192

Reviewer 2 Report
L43: Please provide the full form of the abbreviation EC since it is mentioned for the first time in the manuscript. Same comment for the other abbreviations used for the first time in the text
L68: is the 5% by weight? Please specify.
L73: What does the author mean to communicate by "converging trend"? Please specify the key message.
L75-76: "The BC properties vary widely depending on feedstock material"-please provide brief literature to throw some insight on this point to set the base for the objective of the manuscript.
L78: by "various" do you mean 10, 20, how many different organic materials. Please specify.
L79: as well as its relations with soil carbon and nitrogen content. Include the term soil.
L93: Instead of pH H20 you can use write "pH (aqueous)".
Also, just wondering why total organic carbon is abbreviated as "TOC" while total nitrogen as "Nt" and not "TN".
L125: Methods section is "2.2", Please correct. Secondly, the appropriate term is "analysis" rather than " methods, Section 2.1 is basically the introduction to Section 2, and L95 onwards is the "experimental set up". Please rearrange and use appropriate terms.
L95: was any software for the design of experiments used?
L110-please specify if DI water was used or tap water for the watering.
L118-g/cm3. please check.
L134-Multi N/C 3100. Please correct "Muli".
Fig 1-3: Standard deviation should be shown.
L416: The authors provide the cluster analysis and explain the results of the cluster analysis, however, what is missing is the physical significance of the results related to the objective of the study. For example-"This variant was identified by the highest content of Nt, DOC, DNt, AP as well as activity of DEH, CAT, AlP and AcP"- what is the significance of this statement. PCA analysis is well written, cluster analysis should be similar to that.
Was an attempt made to understand the composition of the raw material? This can provide insight into the biochar properties. In the conclusion, future work could be mentioned. For example, Pyrolysis temperature can be changed and optimized to get the best performing biochar.
Author Response
Dear Reviewer
The paper's authors wish to thank for all the precious comments and guidelines; we have tried to consider all of them.
I hope that the corrections introduced will be satisfactory.
With best regards,
Bozena Debska
Answers:
- L43: Please provide the full form of the abbreviation EC since it is mentioned for the first time in the manuscript. Same comment for the other abbreviations used for the first time in the text
There was: EC
There is: electro conductivity (EC)
There was: CEC – line 45
There is: cation exchange capacity CEC)
- L68: is the 5% by weight? Please specify.
There was: that 5% biochar
There is: that 5 weight percent (wt%) biochar
- L73: What does the author mean to communicate by "converging trend"? Please specify the key message.
There was: … convergent trends
There is: …. the similar trend
- L75-76: "The BC properties vary widely depending on feedstock material"-please provide brief literature to throw some insight on this point to set the base for the objective of the manuscript.
The references were added: [1, 4, 7].
- L78: by "various" do you mean 10, 20, how many different organic materials. Please specify.
There was: from various organic materials
There is: from various organic materials (mellow compost, stabilized municipal sewage sludge, pine sawdust, sycamore sawdust, oak leaves) on …..
- L79: as well as its relations withsoil carbon and nitrogen content. Include the term soil.
There was: …… with carbon and nitrogen content.
There is: ……. with soil carbon and nitrogen content.
- L93: Instead of pH H20 you can use write "pH (aqueous)".
The term pH in H20 is commonly used to describe determination pH in water. Because it is equivalent to the term “pH (aqueous)” we have decided to use this kind description. The description pH H2O was changed to pH in H2O in the manuscript.
- Also, just wondering why total organic carbon is abbreviated as "TOC" while total nitrogen as "Nt" and not "TN".
All symbols “Nt” and “DNt” were corrected to “TN” and “DTN”
- L125: Methods section is "2.2", Please correct. Secondly, the appropriate term is "analysis" rather than " methods, Section 2.1 is basically the introduction to Section 2, and L95 onwards is the "experimental set up". Please rearrange and use appropriate terms.
The number of section 2.2 has been corrected. The title of the section has been changed – there was “Methods”, now there is “Analysis”. The title of section 2.1. was also changed: there was: “Materials”, now there is “Materials and experimental set up”
- L95: was any software for the design of experiments used?
For the evaluation of the experiment design there was not any software used.
- L110-please specify if DI water was used or tap water for the watering.
To keep assumed soil humidity the tap water was used – information added – line 110.
- L118-g/cm3. please check.
Correction done.
- L134-MultiN/C 3100. Please correct "Muli".
Correction done.
- Fig 1-3: Standard deviation should be shown.
The graphs in the fig. 1-3 were corrected. The SD value is now presented.
- L416: The authors provide the cluster analysis and explain the results of the cluster analysis, however, what is missing is the physical significance of the results related to the objective of the study. For example-"This variant was identified by the highest content of Nt, DOC, DNt, AP as well as activity of DEH, CAT, AlP and AcP"- what is the significance of this statement. PCA analysis is well written, cluster analysis should be similar to that.
The greatest Euclidean distance determined for S+MSS variant is related with essential differences of MSS BC properties in comparison to the other biochars. The study of Hossain et al. [47] on waste water sludge biochar indicates that this type of BC presents considerable differences in relation to biochars obtained from other biomass of natural origin [1, 4, 48]
Section added - line 424, two reference position were added [47] and [48]:
[47] Hossain, M.K.; Strezov, V.; Chan, K.Y.; Ziolkowski, A.; Nelson, P.F. Influence of pyrolysis temperature on production and nutrient properties of wastewater sludge biochar. J. Environ. Manage. 2011, 92, 223-228. doi:10.1016/j.jenvman.2010.09.008
[48] Sri Shalini, S.; Palanivelu, K.; Ramachandran, A.; Raghavan, V. Biochar from biomass waste as a renewable carbon material for climate change mitigation in reducing greenhouse gas emissions - a review. Biomass Conv. Bioref. 2021, 11, 2247-2267. https://doi.org/10.1007/s13399-020-00604-5
Was an attempt made to understand the composition of the raw material? This can provide insight into the biochar properties. In the conclusion, future work could be mentioned. For example, Pyrolysis temperature can be changed and optimized to get the best performing biochar.
The raw feedstock material was characterized only by C and N content – basic elements which influence decomposition process in the soil. Being fully aware that one of the most important factors determining the properties of the obtained biochars are: pyrolysis temperature and biomass properties, the properties of biomass (temperature – const.) were selected as the factor determining the BC properties.

Reviewer 3 Report
After reading the manuscript myself I have the following major comments:
1) Why have you compared the biomass carbon from different sources such as agriculture, industry and forestry? If necessary, it should be briefly explained in the introduction.
2) Except for the same amendment amount, how have you ensured the single difference principle in the experimental design?
3) Please explain the reasons for selecting the four enzyme activities. The soil material was sandy loam and pH from natural to acidity (6.4), Why have you determined alkaline phosphatase?
4) Methods about the determination of enzyme activities lack of important details and sometime they are not clear; Did you shake the soil slurry during the determination of dehydrogenases (DEH), catalase (CAT) activity? Shaking soil slurries (soil and solution) is important in all enzyme assays since it facilitates the contact between substrate and enzymes.
5) Due to the considerable variability of soil enzyme activity measured by traditional methods, It is suggested to keep two significant figures after the decimal point in all the tables.
6) All tables are need change into three line tables.
Author Response
Dear Reviewer
The paper's authors wish to thank for all the precious comments and guidelines; we have tried to consider all of them.
I hope that the corrections introduced will be satisfactory.
With best regards,
Bozena Debska
Answers:
The paper has been reviewed and corrected by english nativespeaker. The corrections and introduced changes are indicated in text.
- Why have you compared the biomass carbon from different sources such as agriculture, industry and forestry? If necessary, it should be briefly explained in the introduction.
The biochar used in the study was obtained from municipal sewage sludge, mellow compost, pine and sycamore sawdust and oak leaves. The feedstock material was waste biomass of different origin. Sewage sludge is a waste from municipal waste water treatment plant, mellow compost was produced from “green wastes” and municipal sewage sludge, sawdust is a waste from sawmill. The oak leaves were “green waste” from maintaining park area. Despite the different sources of origin, the common feature of all biochars was the waste status of the biomass used for their production. Information on this issue was added in the line 90.
- Except for the same amendment amount, how have you ensured the single difference principle in the experimental design?
Biochar was mixed with soil in the ratio 1:10 wt without taking into account other BC parameters and properties. The same soil was used in each incubation pot and the incubation conditions were identical for each incubation variant (S+MSS, S+MC, S+SS, S+PS, S+OL).
- Please explain the reasons for selecting the four enzyme activities. The soil material was sandy loam and pH from natural to acidity (6.4), Why have you determined alkaline phosphatase?
Alkaline phosphatase is an enzyme which activity occurs not only in the soil with an alkaline pH (in which it has its optimum). This enzyme has a different origin than acid phosphatase. Both enzymes (alkaline and acid phosphatase) are responsible for the biogeochemistry of soil phosphorus, which was determined in the analyzed soils (available phosphorus).
- Methods about the determination of enzyme activities lack of important details and sometime they are not clear; Did you shake the soil slurry during the determination of dehydrogenases (DEH), catalase (CAT) activity? Shaking soil slurries (soil and solution) is important in all enzyme assays since it facilitates the contact between substrate and enzymes.
In accordance to the enzyme determination procedure used, the soil was shaken twice during the determination of catalase (CAT) [Johnson, J.I.; Temple, K.l. Some variables affecting the measurements of catalase activity in soil. Soil Sci. Soci. Am. 1964, 28(2), 207–209.]. The first time - after adding H2O to the soil (for 15 min). Second time - after adding 0.3% H2O2 (for 20 min).
Also, when determining the activity of denydrogenases (DEH), the soil was properly shaken, according to protocol established by Thalmann, A. Zur methodic derestimung der Dehydrogenaseaktivität und Boden mittels Triphenyltetrazoliumchlorid (TTC). Landwirtsch. Forsch, 1968, 21, 249-258.
- Due to the considerable variability of soil enzyme activity measured by traditional methods, It is suggested to keep two significant figures after the decimal point in all the tables.
As suggested, the values in tables have been corrected.
- All tables are need change into three line tables.
The form of editing tables with the separation of individual lines is, in our opinion, more readable and also used in other publications of the MDPI.
